# Exploring dynamic solvation kinetics at electrocatalyst surfaces

Francisco Sarabia [1,2], Carlos Gomez Rodellar [1,2], Beatriz Roldan Cuenya [1] & Sebastian Z. Oener [1] ✉

The interface between electrocatalyst and electrolyte is highly dynamic. Even in absence of major structural changes, the intermediate coverage and interfacial solvent are bias and time dependent. This is not accounted for in current kinetic models. Here, we study the kinetics of the hydrogen evolution, ammonia oxidation and oxygen reduction reactions on polycrystalline Pt with distinct intrinsic rates and intermediates (e.g. *H, *OH, *NH$_2$, *N). Despite these differences, we discover shared relationships between the pre-exponential factor and the activation energy that we link to solvation kinetics in the presence of electronic excess charge and charged intermediates. Further, we study dynamic changes of these kinetic parameters with a millisecond time resolution during electrosorption and double layer charging and dynamic *N and *NO poisoning. Finally, we discover a pH-dependent activation entropy that explains non-Nernstian overpotential shifts with pH. In sum, our results demonstrate the importance of accounting for a bias and time-dependent interfacial solvent and catalyst surface.

Ions in bulk water are surrounded by an energetically stabilizing shell of water molecules that compensates the large ionization energy of free ions, such as -11.3 eV for H$^{+1}$. This dipolar solvation shell needs to be reorganized whenever ions are passing through the double layer at reactive solid-liquid interfaces across electro- and biochemistry. Despite this broad relevance, ion solvation kinetics are poorly understood, limiting fundamental understanding and development of new technology. For example, currently, it is unclear whether new electrolyte design could selectively tailor the solvation kinetics of specific ions to enhance catalyst activity, selectivity and stability.

In electrochemistry, outer-sphere Marcus theory has historically been used to capture the effect of solvent reorganization solely in the reaction's enthalpy. Here it is assumed that the solvent dielectric environment outside of the coordination shell of a redox molecule does not drastically change when the molecule's charged state is altered upon electron tunneling through the double layer[1]. In contrast, when an ion is passing through the inner-sphere of the electrochemical double layer, the ion is exposed to a heterogenous electrostatic environment with a non-trivial bias dependence. In this complex dielectric environment, the ion needs to shed/ reorganize its solvation shell. As a result, a priori the bias dependent solvent reorganization could also impact the activation entropy. Conversely, there is no rigorous theoretical basis[2] for the validity of outer-sphere Butler-Volmer/ Marcus-Hush-type theories, that assume constant activation entropies and pre-exponential factors for inner-sphere reactivity in aqueous media[3,4].

For electrocatalysis, a priori only a general Arrhenius rate law with a variable activation energy and pre-exponential factor is valid. According to statistical mechanics and transition state theory[5,6], the pre-exponential factor contains the (activation) entropy difference ($\Delta S$) between the transition and initial state. For a solid-liquid interface, this includes changes of the configurational entropy upon surface coverage with reaction intermediates and changes in the entropy of interfacial solvent dipoles between the initial and the transition state. At applied bias, all of these entropic contributions might change, which has challenged the interpretation of potential dependent pre-exponential factors in electrocatalysis[4,7–9]. While a bias dependent surface coverage of reaction intermediates could, in principle, inform

[1]Department of Interface Science, Fritz-Haber Institute of the Max Planck Society, Berlin, Germany. [2]These authors contributed equally: Francisco Sarabia, Carlos Gomez Rodellar. ✉e-mail: oener@fhi-berlin.mpg.de

on surface configurational entropy changes, the impact of electric fields on the solvent side has remained particularly elusive. This has limited understanding, not only in electrocatalysis, but also about the role of electric fields in biocatalysis[10,11]. This is due to (i) the difficulty of capturing surface excess charge and long-range[12] solvent dynamics with density functional theory (DFT) and the (traditionally) large computational cost of molecular dynamics (MD)[13,14], (ii) the challenges of *operando* spectroscopic characterization of solvents at reactive interfaces[15] and (iii) the scarcity of temperature dependent kinetic data sets[4], that could, in fact, inform on changes in the activation entropy and enthalpy with bias.

Here, we determine the activation entropy and enthalpy for three different Faradaic reactions and various pseudo-capacitive electrosorption processes across a wide potential range ($-0.1 - 1.0$ V$_{RHE}$) on the same polycrystalline Pt surface in alkaline media. We discover strikingly similar relationships in the kinetics that we interpret as the common fingerprints of interfacial solvation in alkaline media. To delineate entropic changes on the surface from the ones in the interfacial solvent, we temporally resolve the dynamic coverage with *N and *NO intermediates that poison the Pt surface during the electrocatalytic oxidation of the $NH_3$ molecule. Finally, we show that a pH dependent activation entropy can cause non-Nernstian overpotential shifts with pH in electrocatalysis.

## Results

The hydrogen evolution reaction (HER, $2H_2O + 2e^- \rightarrow H_2 + 2OH^-$), ammonia oxidation reaction (AOR, $6OH^- + 2NH_3 \rightarrow N_2 + 6H_2O + 6e^-$) and oxygen reduction reaction (ORR, $O_2 + 4e^- + 2H_2O \rightarrow 4OH^-$) on polycrystalline Pt in alkaline media (0.1 M KOH) are key reactions for the generation and utilization of green hydrogen and green ammonia, but are penalized by high kinetic overpotentials (Fig. 1a,b). Due to the vastly different reaction pathways and intermediates on the Pt surface, such as *H, *OH, *O, *NH$_2$ and *N, the reaction kinetics are predicted to be largely independent by traditional catalyst activity descriptors, such as the intermediate's binding energy. Despite these differences, we discover shared fingerprints in the kinetics for all of these processes when we extract the bias dependent pre-exponential factor ($A(\eta)$) and

activation energy ($E_A(\eta)$) from temperature dependent studies and Arrhenius fits with $R^2 > 0.9$ (Supplementary Fig. 1, Methods). Note, for the Arrhenius analysis of the Faradaic reactions, we perform multistep chronoamperometry with individual potential holds of 15 s (see Methods for details). Conversely, the extracted $A$ and $E_A$ inform on the rate-limiting step of the Faradaic reactions and not on (pseudo-)capacitive processes. The analysis of pseudo-capacitive (dis)charge and deactivation kinetics will be discussed below.

The HER, ORR and AOR all exhibit an extended potential range for which $\log A(\eta)$ and $E_A(\eta)$ are linearly dependent on each other ($R^2 - 0.99$) with a slope of $\Delta \log A(\eta) \cdot \Delta E_A(\eta)^{-1} \sim 0.19 - 0.22$ mol kJ$^{-1}$ (Fig. 1c). Importantly, $E_A(\eta)$ increases with bias and the entropic changes in $A(\eta)$ are driving the increasing rates with bias. This behavior contrasts Butler-Volmer and Marcus-Hush-type theories that predict increasing rates only with decreasing activation energies. While the absolute values of $E_A(\eta)$ and $A(\eta)$ are different between the reactions, the linear compensation slopes ($\Delta \log A(\eta) \cdot \Delta E_A(\eta)^{-1}$) are similar in magnitude, indicative of a shared origin. For the HER, the slope is further independent of the $H_2$ concentration in the electrolyte (Supplementary Fig. 2).

Recently, we studied isolated water dissociation and hydroxide solvation kinetics in bipolar membranes and at metal-electrolyte interfaces, including for the HER on polycrystalline PdAg foils[16], and observed slopes of $\Delta \log A(\eta) \cdot \Delta E_A(\eta)^{-1} \sim 0.25$ mol kJ$^{-1}$ across the chemically dissimilar polymer and metal interfaces. Thus, we conclude that the similar slope for the HER on Pt in Fig. 1c (0.22 mol kJ$^{-1}$) is directly linked to water dissociation and hydroxide solvation kinetics at the Pt surface, which limits the HER in alkaline conditions ($2H_2O + 2e^- \rightarrow H_2 + 2OH^-$)[17]. Previously, increasing excess charge and electric fields have been linked to slow hydroxide solvation kinetics for the HER in alkaline conditions[18]. The results in Fig. 1c demonstrate that $E_A(\eta)$ indeed increases with polarization, but the bias dependent activation entropy increases, too, and, in fact, is responsible for the increasing rates. Conversely, a more ordered hydrogen bond network appears to promote the interfacial charge transfer and does not suppress it, as commonly assumed in the literature.

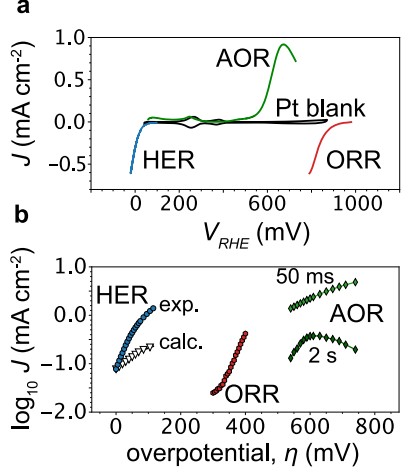

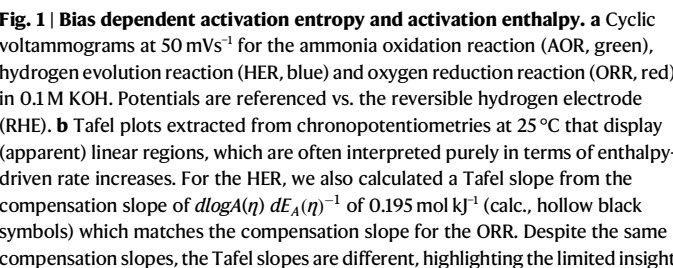

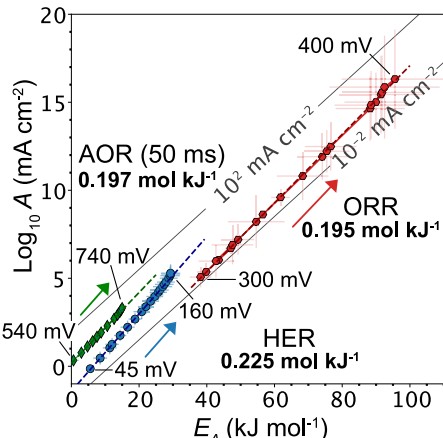

**Fig. 1 | Bias dependent activation entropy and activation enthalpy. a** Cyclic voltammograms at 50 mVs$^{-1}$ for the ammonia oxidation reaction (AOR, green), hydrogen evolution reaction (HER, blue) and oxygen reduction reaction (ORR, red) in 0.1 M KOH. Potentials are referenced vs. the reversible hydrogen electrode (RHE). **b** Tafel plots extracted from chronopotentiometries at 25 °C that display (apparent) linear regions, which are often interpreted purely in terms of enthalpy-driven rate increases. For the HER, we also calculated a Tafel slope from the compensation slope of $d\log A(\eta)\,dE_A(\eta)^{-1}$ of 0.195 mol kJ$^{-1}$ (calc., hollow black symbols) which matches the compensation slope for the ORR. Despite the same compensation slopes, the Tafel slopes are different, highlighting the limited insight

provided by Tafel analysis at one temperature. **c** Overpotential ($\eta$) dependent pre-exponential factor, $A(\eta)$, and activation energy, $E_A(\eta)$, for the HER (blue), ORR (red) and AOR at 50 ms (green) (longer times below). Contrary to common belief, the activation energy, in fact, increases with bias at low current densities in alkaline media. Despite different reaction intermediates and pathways, all reactions display very similar slopes of $d\log A(\eta)\,dE_A(\eta)^{-1}$. $E_A$ and $A$ values are obtained from the slope and origin intercept of the Arrhenius linear regression and based on four-five observations (temperatures). Error bars are extracted from the standard errors. All $R^2$ values are shown in Supplementary Fig. 1.

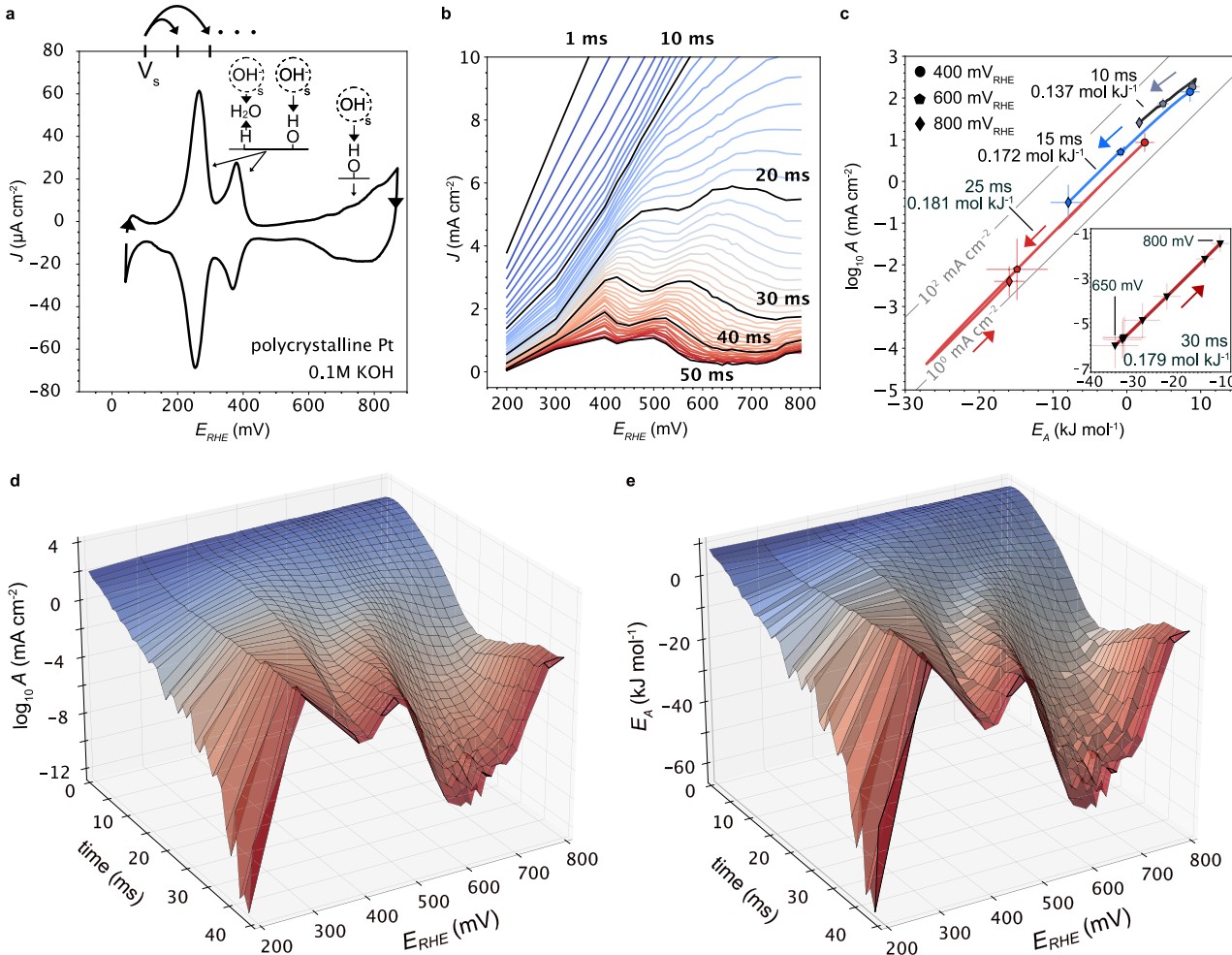

**Fig. 2 | Pseudo-capacitive discharge kinetics on polycrystalline Pt. a** Cyclic voltammogram (50 mVs⁻¹) of a polycrystalline Pt foil in 0.1 M KOH. Temperature dependent potential jumps were performed between the starting potential ($V_S$) and positive potentials up to 800 mV$_{RHE}$. In anodic direction, the electrosorption processes include H desorption, OH adsorption and the onset of Pt oxidation, all of which involve water formation and hydroxide desolvation, as indicated. **b** Potential dependent (dis)charge current transients after the potential jumps. After ~10 ms, the discharge currents start to resolve the potential dependent (dis)charge peaks, including the onset of Pt oxidation at higher potentials. **c** Bias dependent pre-exponential factor (log$A(\eta)$) and activation energy ($E_A(\eta)$) for (dis)charge currents at different times after the potential jumps (Supplementary Fig. 3). We observe a decreasing log$A(\eta)$ and $E_A(\eta)$, since higher temperatures ($T$) lead to lower currents (J) ($E_A \propto -d\log J dT < 0$). Note, here, a lower slope dlog$A(\eta)E_A(\eta)^{-1}$ represents faster discharge kinetics, in contrast to the kinetics at Faradaic currents. At longer times

and higher potentials, we observe a reversal of the slope for the onset of Pt oxidation, where currents increase with temperature. For a full video with 1 ms time resolution see Supplementary Movie 1. $E_A$ and $A$ are means and error bars show standard deviation for $E_A$ (slope) and $A$ (intercept) from Arrhenius analysis, based on five observations (temperatures). **d**, **e** Time dependence of log$A(\eta)$ and $E_A(\eta)$, respectively. After 10–20 ms, the electrosorption peaks in Fig. 2a appear. These potential dependent variations around 300–600 mV are not apparent in panel c, as log$A(\eta)$ and $E_A(\eta)$ compensate along the same line. The bias dependent compensation of log$A(\eta)$ and $E_A(\eta)$ in (**d**, **e**) is likely related to the impact of bias dependent build-up of excess charge at local surface motifs on the solvation kinetics. For (**d**, **e**) $E_A$ and $A$ are means. Error bars are not shown for clarity. $R^2$ values based on five observations are shown in Supplementary Fig. 4. For more discussion on the linear fit see Supplementary Note 2. All potentials are referenced vs. the reversible hydrogen electrode (RHE).

Large configurational entropy changes are critical during bulk water auto dissociation[19] and even proton and hydroxide transport in the bulk. Grotthuss transport is mediated by fluctuating hydrogen bond reorganization and follows Arrhenius behavior with a low activation enthalpy (∼ 10.5 kJ mol⁻¹). However, whenever H-bond fluctuations are constrained, e.g. due to high halide counter ion concentration, proton transport can be limited by a low activation entropy[20]. Conversely, when moving to the inner-sphere of a heterogeneous interface, the constrained degrees of freedom of the water network can impact the transfer of ions through the double layer. This is particularly true, whenever excess charge emerges at the solid-liquid interface.

Interfacial excess charge can impact both, the activation entropy and enthalpy, which we have previously linked[16] to the presence of

compensation slopes $\Delta \log A(\eta) \cdot \Delta E_A(\eta)^{-1}$. On the one hand, excess charge leads to an increasing electric field that impacts the activation entropy of the ion solvation step. More specifically, we hypothesize that increasing electric fields increase the ion's configurational transition state entropy, $S_t$, due to spatial delocalization of the ion in the ordered water network. Additionally, the electric field might also reduce the entropy of the reactant $H_2O$ molecule, $S_i$. The result is an increasing activation entropy, $\Delta S = S_t - S_i$. On the other hand, excess charge at the active site is energetically penalized by increasing activation energies, which has theoretically been predicted, e.g., for hydrogen desorption and proton solvation at Au surfaces[1] and inferred from experimental results on the impact of the difference between the onset potential of a Faradaic reaction and potential of zero charge[18]. Importantly, the effect of excess charge and electric fields on the

activation entropy has not been addressed in these studies, due to the large computational costs of capturing extended water ordering and missing temperature dependent data sets.

## The kinetic fingerprints of charged intermediates

For the ORR and AOR, we obtain lower slopes ($\sim 0.19$ mol kJ$^{-1}$), that are similar to the slopes we obtain for pseudo-capacitive Pt oxidation ($\sim 0.18$ mol kJ$^{-1}$) (more below). Thus, we tentatively ascribe the lower slopes for the AOR and ORR to the formation of charged intermediates in a potential range that is paralleled by Pt oxidation. To incrementally increase $A(\eta)$ via excess charge, more bias is needed due to parallel Pt oxidation that reduces some of the excess charge density ($Pt - OH + e^- \leftrightarrow Pt + OH^-$), compared to a fully metallic surface. Importantly, in contrast to electronic polarization for the HER, for the AOR and ORR, excess charge is likely linked to charged intermediates. For the AOR, a rate-limiting NH$^-$ intermediate has been predicted by DFT (more below)[21,22], while the superoxide O$_2^-$ intermediate is considered the limiting ORR intermediate in alkali and has been detected by attenuated total reflection infrared spectroscopy on polycrystalline Pt[23]. Taken together, our results strongly suggest that the bias (and time) dependent activation enthalpy and entropy, such as in Fig. 1c, directly inform on the generation of excess charge at the interface. Depending on the reaction, excess charge might be needed to (de) solvate ions, polarize and break strong chemical bonds or bind molecules to the surface. This is critical for inner-sphere reactivity at electrocatalyst interfaces and strongly contrasts the role of excess charge in (non-catalytic) outer-sphere reactions. As mentioned above, changes in the charged state of the redox molecule upon electron tunneling lead, primarily, to changes in the activation enthalpy, while leaving the activation entropy unaffected. Conversely, the resulting Butler-Volmer and Marcus-Hush-type kinetics are typically analyzed via Tafel analysis at a constant temperature.

The Tafel plots for the HER and ORR display apparent linear regions (Fig. 1b), despite non-constant pre-exponential factors and increasing activation energies with bias (Fig. 1c). This exemplifies how, for inner-sphere electrocatalyst activity, linear Tafel slopes at one temperature are insufficient to ascribe increasing rates a priori solely to changes in the activation enthalpy, let alone inform on the number of electrons involved in any reaction steps[24]. To highlight the limitations of traditional Tafel analysis at a single temperature, we simulated the HER kinetics with a $dlogA(\eta)dE_A(\eta)^{-1}$ slope of 0.195 mol kJ$^{-1}$ (hollow black symbols in Fig. 1b) leading to a lower Tafel slope than the ORR, albeit the same $dlogA(\eta)dE_A(\eta)^{-1}$ (red line in Fig. 1c). To overcome these constraints, temperature dependent Tafel slopes inform on the susceptibilities of the activation enthalpy ($\alpha_H$) and entropy ($\alpha_S$) to the electrochemical bias, as outlined by Conway[7,8,25,26]. However, for reliable analysis, this approach requires mass-transport free regions with linear Tafel slopes over decades of current density. Thus, we analyze $A(\eta)$ and $E_A(\eta)$ directly, including at low overpotential, where Tafel approximation is limited.

## Bridging electrosorption and electrocatalyst kinetics

The link between pseudo-capacitive (dis)charge and Faradaic currents is critical to bridge fundamental electrochemistry relying on cyclic voltammetry with electrocatalyst investigations at Faradaic turn-over. For example, Kuo et al. studied different electro-adsorption rates of surface-bound H on Pt(111) with scan rates of up to 1000 Vs$^{-1}$ in acid and base and concluded that interfacial water ordering might be the underlying reason for the slow rates in base[27]. Recently, Lewis et al. used scan rates of up to 1000 Vs$^{-1}$ to study the pseudo-capacitive (de) protonation of graphite-conjugated carboxylic acids ($GC - COOH \leftrightarrow GC - COO^- + H^+$)[28] and obtained a fast rate constant in acid and a slow one involving water dissociation/formation in base. These studies demonstrate differences in the kinetics between acid in base. However, they also raise questions about the link between

double layer charging and electrosorption kinetics. Therefore, we perform temperature dependent potential jump measurements to inform on pseudo-capacitive (dis)charge kinetics.

Figure 2a shows the blank cyclic voltammogram of the polycrystalline Pt foil with well-known pseudo-capacitive peaks that are associated with H$^+$ and OH$^-$ ad- and desorption ($0.1-0.4$ V$_{RHE}$) and the onset of OH adsorption and Pt oxidation (>0.55 V$_{RHE}$). As indicated in the figure, we performed potential jumps between a resting potential at 0.1 V$_{RHE}$ and potentials between 0.2 and 0.8 V$_{RHE}$. Figure 2b shows the pseudo-capacitive discharge and charging currents at 25 °C, resolving the evolution of the two H$^+$, OH$^-$ ad- and desorption peaks after 10 ms and the onset of Pt oxidation after 20 ms. These studies were repeated at different temperatures (Supplementary Fig. 3) and Arrhenius analysis applied to extract E$_A$ and A with a time resolution of 1 ms, with snapshots shown in Fig. 2c. The dynamic (dis)charge kinetics are best visualized in Supplementary Video 1.

The Arrhenius equation does not involve a time parameter and, thus, is only strictly valid at steady-state (e.g. Fig. 1c). However, we observe the evolution of the current peaks in Fig. 2b that match the peaks in the cyclic voltammogram in Fig. 2a and which have been identified with the different electrosorption processes in the literature previously. Conversely, while the currents in Fig. 2b are stemming from a combination of pseudo-capacitive electrosorption and double layer charging at the Pt surface, the (relative) evolution of the peaks at times > 10 ms can be identified with the respective electrosorption processes (more below). We specifically note, that the time constants observed here are not the time constants of the interfacial charge transfer itself[29]. The time evolution here is related to the separation of double layer (dis)charging, which can take ms (depending on cell dimensions[30]) from electrosorption currents (Supplementary Note 1). The latter can be described with an Arrhenius rate law.

With increasing temperatures (T), the pseudo-capacitive discharge currents (J) in Fig. 2b are lower for a given time point (faster capacitive discharge with increasing temperatures). This results in negative activation energies ($E_A \propto d \log J dT < 0$) and a slope of $\sim 0.13$ mol kJ$^{-1}$ (10 ms) to $\sim 0.18$ mol kJ$^{-1}$ (25 ms) in Fig. 2c that moves in opposite direction to the ones of pseudo-capacitive charging currents at the onset of Pt oxidation (>650 mV$_{RHE}$) and the ones of the Faradaic currents. There, an increasing temperature leads to increasing rates. Strikingly, for any given time, $E_A$ and A compensate along the same slope for all potentials, even during the evolution of the electrosorption peaks in Fig. 2b. At higher times (inset of Fig. 2c), we observe a sudden trend reversal and a slope of $\sim 0.18$ mol kJ$^{-1}$ for potentials that correspond to the onset of Pt oxidation, where pseudo-capacitive charging currents increase with temperature. Figure 2d, e show the time evolution of $E_A$ and A for different potentials to better visualize the correlation with the electrosorption peaks in Fig. 2a, for which $E_A$ and A locally increase. As noted above, the absolute values of A and $E_A$ are stemming from transient currents (Fig. 2c) that arise due to electrosorption and double layer charging. However, the relative changes in A and $E_A$ with bias closely resemble to electrosorption peaks in the cyclic voltammogram in Fig. 2a, starting at times > 10 ms. Besides the double layer charging, the potential jump might induce a local pH swing, especially at shorter times. However, even in presence of the latter, the current transients in Fig. 2b clearly resolve the pseudo-capacitive peaks in the cyclic voltammograms, which are the ones used to extract $E_A$ and A (Fig. 2c–e). Furthermore, at longer times (≥25 ms), the slopes in Fig. 2c slowly approach the steady-state values for the AOR and ORR (Fig. 1c). Conversely, a local pH swing might impact the solvation kinetics compared to low steady-state currents, but it does not change the general interpretation of our results.

Taken together, the results in Fig. 2 demonstrate that (dis)charge kinetics, which involve electrosorption and double layer charging, are fundamentally governed by similar log$A$ - $E_A$ compensation slopes as for Faradaic reactions in alkaline media (Fig. 1c, d). This finding is

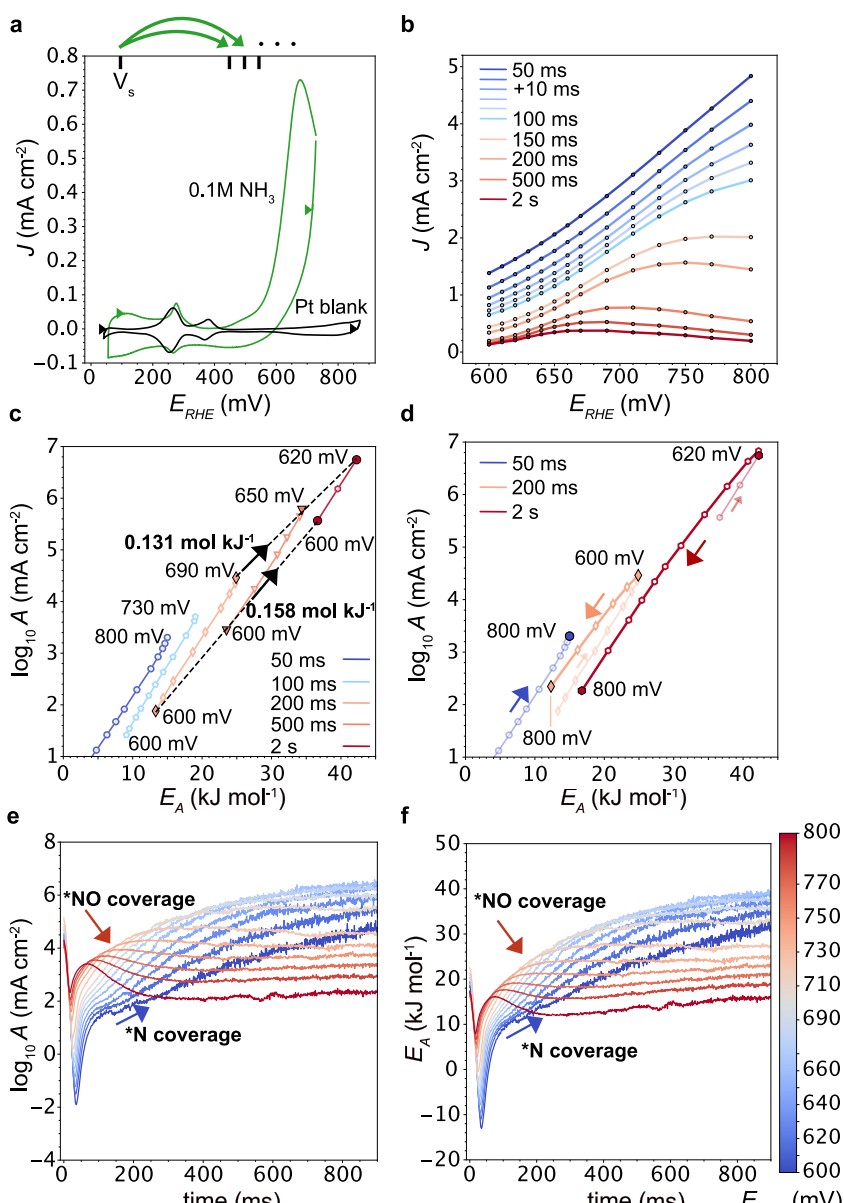

**Fig. 3 | Temporally resolved ammonia oxidation kinetics. a** Cyclic voltammogram (50 mVs⁻¹) of a polycrystalline Pt foil in 0.1 M KOH (Pt blank) and with added 0.1 M NH₃. Temperature dependent potential jumps were performed between the starting potential, $V_s$ = 0.1 $V_{RHE}$, and positive potentials up to 0.8 $V_{RHE}$ (Supplementary Fig. 10). **b** Voltammograms extracted from potential jump measurements display rapid deactivation. For short times, the voltammograms are offset by a background current that originates from a mix of capacitive discharge and Faradaic currents. **c** log$A$ and $E_A$ as a function of time and potential. With time, increasing coverage with *N intermediates competes increasingly with primary AOR intermediates, leading to increasing log$A$ and $E_A$. The slopes at constant potential, $\Delta \log A_{600mV}(\tau) \cdot \Delta E_{A,600mV}(\tau)^{-1} \sim 0.16$ mol kJ⁻¹, or the potential of maximum A and $E_A$, $\Delta \log A_{max}(\tau) \cdot \Delta E_{A,max}(\tau)^{-1} \sim 0.137$ mol kJ⁻¹ (black dotted lines), are below the iso-current lines (0.175 mol kJ⁻¹ at 25 °C) and the bias dependent slopes

(0.19 − 0.22 mol kJ⁻¹). Thus, we assign the trends at constant potentials (black dashed lines) to a compensation effect between the surface configurational entropy and activation energy that leads to an effective *N deactivation of the surface. **d** log$A$ and $E_A$ for higher potentials and longer times, showing the effect of NO deactivation. In line with our results in Fig. 2, we interpret the sudden trend reversal and negative slopes as the effect of capacitive discharge of the surface due to Pt-NO or Pt-ON formation. **e-f** $\Delta \log A(\eta, \tau)$ and $\Delta E_A(\eta, \tau)$ with 1 ms resolution, respectively. The *N poisoning at short times shows a distinctly different impact on the AOR kinetics than the *NO poisoning at longer times. The dynamic nature of the kinetics is much more apparent in the Supplementary Movie 2. All potentials are referenced vs. the reversible hydrogen electrode (RHE). Only Arrhenius data with $R^2 > 0.9$ is shown. The error bars are omitted for clarity, but can be found in Supplementary Fig. 11.

critical to bridge fundamental electrochemistry findings extracted from cyclic voltammograms, with studies on electrocatalyst kinetics. In particular, the log$A$ - $E_A$ compensation slopes imply that excess charge localized at surface motifs (crystal facets, step edges etc.) is likely critical to for electrosorption kinetics. We note that even during the evolution of the characteristic peak shapes in log$A$ and $E_A$ (Fig. 2d, e), the transient discharge kinetics involve ion (de)solvation during electrosorption and water dipole reorientation during double

layer charging. In fact, we hypothesize that ion (de)solvation in a dynamically changing dipole water network might be accelerated, possibly giving rise to the different compensation slopes at shorter times (≤25 ms) in Fig. 2c. This would imply that slow relaxation time constants in the ordered water network (Supplementary Note 1) can be modified via potential pulsing, which is used regularly during electrodeposition or for electrocatalytic reactions. After longer times, we observe that the changes in log$A$ and $E_A$ approach the values for the

steady-state Faradaic reactions, that are essentially devoid of transient double layer charging.

Electrosorption kinetics have previously been studied with (high frequency) impedance[30–32], where some studies concluded that H/OH electrosorption rates are too fast to be able to detect the charge transfer resistance associated with electrosorption[30,31], while others isolated it for slow hydrogen underpotential deposition in alkali[32]. These latter findings are also consistent with high-scan rate cyclic voltammetry studies, that studied electrosorption kinetics[27,28]. However, the role of the activation entropy and enthalpy did not emerge from these studies. Thus, to better understand the link between capacitance and electrosorption kinetics, we performed impedance measurements at HER and ORR potentials. For the AOR, deactivation with *N and *NO prevents reliable impedance spectroscopy. For details about the impedance measurements, see Supplementary Note 3 and Supplementary Figs. 5–7. For the HER and ORR, we find linear correlations ($R^2 \geq 0.96$) in the changes of the real capacitance, $|\Delta C_R(\nu,\eta)|$, with $\log_{10}\Delta A(\eta)$ and $\Delta E_A(\eta)$ across a large frequency range and shown at ~126 Hz and ~63 Hz as representative cases in Supplementary Fig. 8. These frequencies match the time-constants observed in Fig. 2, where we see the emergence of electrosorption currents after ~10 ms after the potential jump. The results support that the capacitance changes are linked to electrosorption and double layer charging/ relaxation. In the literature, the latter are often associated falsely with mass transport limitations or local pH changes (Supplementary Note 4).

Electrolyte ions are a critical in electrochemistry[33,34] and have been extensively shown to impact interfacial capacitance and catalyst kinetics. Previously, it has been hypothesized that anion adsorption during the HER in acid could lead to increasing pre-exponential factors, because it could draw the charged protons to the surface and, thus, increase the reactant concentration[4]. However, for the alkaline HER kinetics in Fig. 1, the reactant is neutral water and we observe a compensation slope of $\Delta \log A(\eta) \cdot \Delta E_A(\eta)^{-1} \sim 0.22$ mol kJ$^{-1}$, which is similar to the ones we observed for pure water humified bipolar membranes (0.25 mol kJ$^{-1}$)[16] in absence of auxiliary electrolyte. Therefore, we assign the entropic changes in Figs. 1–2 primarily to the intrinsic nature of interfacial water dipole ordering, but suspect that electrolyte ions can modify $\Delta S$ and $\Delta H$. This will especially be true, if electrolyte ions adsorb to the surface, impact the hydrogen bond network or are needed to form the activated complex, such as during $CO_2$ reduction[35].

To summarize, the following observations support the assignment of the entropic changes in Figs. 1–2 primarily to the solvent side; we observe similar slopes across vastly different reactions on polycrystalline Pt (Fig.1) that are furthermore similar to compensation slopes we observed previously for chemically dissimilar polymer and PdAg-KOH interfaces[16]. Additionally, for the individual snap shots in Fig. 2c, the compensation slope is constant across the whole potential range and independent of the cyclic voltammogram peaks. If surface configurational entropy changes were impacting the compensation slopes, the latter should change their values during the evolution of the cyclic voltammogram peaks. For example, for the discharge current at ~250 mV$_{RHE}$ (Fig. 2a), the surface coverage changes more strongly than for the peak around 350 mV$_{RHE}$, as it is the case for the onset of Pt oxidation. Nonetheless, the compensation slope stays the same (at a fixed time). Additionally, to reconcile Figs. 1–2 with surface configurational entropy changes alone, a universal coverage dependence for vastly different intermediates and catalyst surfaces would have to be postulated. In particular, increasing $E_A$ with bias would imply a universal decrease of coverage of primary intermediates with bias.

Besides surface configurational entropy changes, the reactant concentration in the pre-exponential factor could also change with bias. Previously, for proton desolvation during the HER, it has been hypothesized that compensation between $E_A$ and log A might occur

due to a bias dependent concentration changes of $H_3O^+$ and a reduction of the distance to the electrode[4]. However, here and in our previous study[16], the reactants includes water molecules at a high concentration at metal(oxide) and polymer sites, as well as $NH_3$ and $O_2$ molecules. Also, an impact of a bias dependent reactant concentration should result in bias dependent slopes, in contrast to our results in Fig. 2c. Finally, we want to stress that the absolute values of $\log A$ and $E_A$ are different for different reactions (e.g. Fig. 1), and likely strongly impacted by absolute surface configurational entropy and the binding energy of intermediates. So far, our analysis has primarily focused on the bias dependent changes in $\log A$ and $E_A$, i.e. the compensation slopes at the onset regions of all reactions. Next, we turn to surface configurational entropy.

## Surface configurational entropy-enthalpy compensation

Delineating surface configurational entropy from entropic changes in the interfacial solvent is critical in analyzing bias dependent pre-exponential factors. In gas phase catalysis and, thus, in absence of a solvent and (apparent) electric fields, Constable-Cremer compensation between surface configurational entropy and activation energy were proposed, as both are linked to the coverage of intermediates[36–38]. Supplementary Fig. 9 shows configurational changes as a function of coverage, with steep increases in configurational entropy changes only for strongly covered or almost empty surfaces. In electrocatalysis, isolating surface configurational entropy is very challenging for interfaces exposed to bias dependent electric fields and intermediate coverage, as discussed by Zeradjanin and coworkers[39–41]. To the best of our knowledge, there exists no experimental data on isolated configurational entropy changes in electrocatalysis.

To isolate the influence of surface configurational entropy, we performed potential jump measurements in the presence of $NH_3$ and recorded the AOR current ($6OH^- + 2NH_3 \rightarrow N_2 + 6H_2O + 6e^-$). Initially, *N was thought to be needed for $N_2$ evolution[42], before Gerischer and Maurer[43] concluded that the high *N binding energy must lead to deactivation with increasing coverage, and the AOR proceeds instead over $N_xH_y$ intermediates. At larger potentials $\geq 0.65$ V, the Pt surface is characterized by the onset of oxidation (see Fig. 2a for a Pt blank voltammogram and Fig. 2c for the kinetics) and additional poisoning with NO has been observed by *operando* ATR-IR by Matsui et al.[44]. As a result, we chose the AOR to study the time-dependent coverage of strongly binding N and NO surface species on the catalyst kinetics.

Figure 3a shows cyclic voltammograms of the polycrystalline Pt foil in 0.1 M KOH in absence (black) and presence (green) of 0.1 M $NH_3$ and the indicated jumps from the resting potential ($V_S = 0.1V_{RHE}$) to onset potentials of Faradaic AOR current. Figure 3b displays the effect of the deactivation in the voltammograms starting at 40 ms after the jump, emphasizing the limitations of regular cyclic voltammetry on highly dynamic surfaces. For times <40 ms, we observe capacitive discharge currents, similar to the results in Fig. 2 (Supplementary Fig. 12). Figure 3c shows that log A and $E_A$ at a constant bias parallel shift diagonally to higher values (black dashed lines) with slopes $\Delta \log A_{600mV}(\tau) \cdot \Delta E_{A,600mV}(\tau)^{-1} \sim 0.16$ mol kJ$^{-1}$ and $\Delta \log A_{max}(\tau) \cdot \Delta E_{A,max}(\tau)^{-1} \sim 0.13$ mol kJ$^{-1}$. We ascribe the parallel shifting lines to successive N-coverage and blocking of active-sites. This leads to a reduction in primary AOR intermediates, which manifests itself in an increase in $E_A$ and a surface configurational entropy-driven increase in $\log A$. Importantly, this slope is lower than the one of the iso-current line (~0.175 mol kJ$^{-1}$ at 25 °C), hence, it leads to deactivation. The slopes causing the increasing rates with bias are substantially larger (0.19 − 0.22 mol kJ$^{-1}$). Figure 3d shows log A and $E_A$ for $\geq 0.65$ V where additional poisoning with NO decreases the AOR rates even more strongly, finally resulting in complete deactivation. Figure 3e, f show $E_A$ and log A with a 1 ms time resolution. The dynamic nature of the kinetics becomes much more apparent in Supplementary Movie 2.

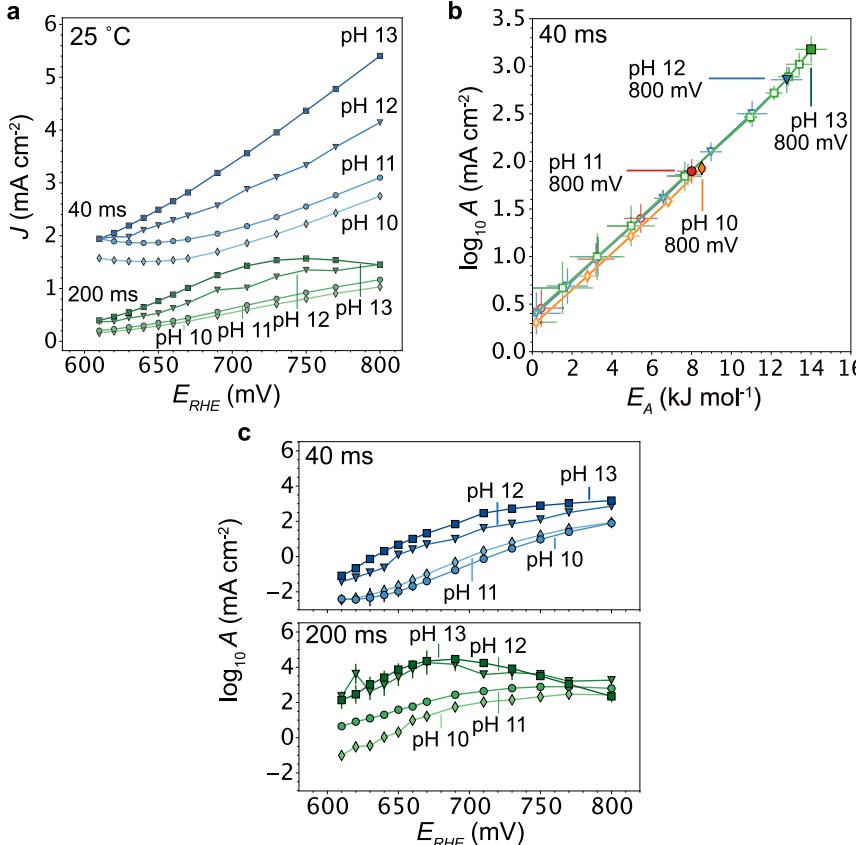

**Fig. 4 | pH dependent activation entropy and non-Nernstian AOR kinetics.**
**a** Voltammograms extracted from potential jump measurements show decreasing AOR activity with pH referenced vs. the reversible hydrogen electrode (RHE). At 40 ms, the voltammograms are offset by a background current that originates from a mix of capacitive discharge and Faradaic currents. **b** Kinetic log $A$ vs. $E_A$ maps for the AOR at different pH at 40 ms (for 200 ms see Supplementary Fig. 13). With decreasing pH, the pre-exponential factor decreases (at the same overpotential) and leads to decreasing AOR activity, despite the decreasing activation energy. The compensation slope stays constant, $d \log A(\eta) dE_A^{-1} = 0.183 - 0.19$ mol kJ$^{-1}$. **c** At lower pH, the pre-exponential factor never reaches the maximum obtained at high pH. $E_A$ and $A$ values are means and error bars are based on five observations (temperatures) with all $R^2$ values shown in Supplementary Fig. 14 (see also Supplementary Note 2). For full time resolution of the pH dependent kinetics, see Supplementary Movies 2-5.

Taken together, the data in Fig. 3 temporally resolves the different impact of N- and NO-poisoning on the AOR mechanism and link it to surface configurational entropy changes. N-species form even at low potentials and early times and compete with AOR intermediates for active sites. In contrast, NO formation likely occurs only substantially at the onset for PtOx formation (≥0.65 V). At these potentials, the Pt surface is sufficiently charged to readily enable hydroxide desolvation (see Fig. 2a, c for the kinetics of $Pt^+ + OH^- \rightarrow PtOH$). As a result, additional generation of Pt-NO (or Pt-ON) might capacitively discharge (or electronically passivate) the Pt surface and the $d \log A(\eta) dE_A^{-1}$ slope reverses and changes sign, mirroring the trend-reversal observed in Fig. 2 for pseudo-capacitive discharge. These results highlight the importance of considering configurational entropy changes of the surface and the interfacial solvent to understand catalyst deactivation. Additionally, they strongly motivate efforts for increased time resolution and surface sensitivity of *operando* spectroscopic characterization.

**Non-Nernstian kinetics and pH-dependent activation entropy**
To reconcile non-Nernstian overpotential shifts with changing electrolyte pH, previous electrocatalyst studies have often proposed pH-dependent surface charge, related to the difference of the onset potential of a Faradaic reaction[18] and the potential of zero total charge or the formation of (charged) intermediates via proton-decoupled electron transfer steps. For example, a charged AOR intermediate on Pt has been theoretically predicted by DFT based on the strong binding

of NH$_x$[22,45,46]. The electron-decoupled hydroxide charge transfer is thought to lead to a direct pH dependence in the formation of the intermediate $(OH^- + {}^*NH_2 \rightarrow (\mathbf{NH})^- + H_2O)$. However, as discussed above, these studies did not consider the effect of the pH on the activation entropy and the results are generally solely interpreted according to theoretical models, where the intermediate's binding energy is the primary activity descriptor. This approach neglects the role of the (explicit) solvent and we, thus, hypothesized that charged intermediates electrostatically induce changes in the entropy of the interfacial solvent.

Figure 4a shows AOR polarization curves that have been extracted from the potential jump measurements at 25 °C and different pH. With decreasing pH, the activity on the reversible hydrogen electrode (RHE) potential scale is substantially reduced, but also the apparent *NO deactivation at longer times and higher overpotentials. Strikingly, when we extract the kinetic maps (Fig. 4b), we observe *decreasing* activation energies with decreasing pH, while the slope $d \log A dE_A^{-1}$ stays constant. Thus, to understand decreasing activity with pH, the pH dependent pre-exponential factor is, yet again, critical at all potentials and times (Fig. 4c), whereas the activation energy in isolation would predict the exact opposite activity trend with pH. To reach the same activation entropy, more bias is required at a low pH. However, for lower pH values, even at higher potentials the activation entropy never reaches the values obtained at higher pH. At potentials ≥670 mV, a pH effect on the NO deactivation is apparent, which we tentatively ascribe to another pH dependent step in the NO formation or capacitive

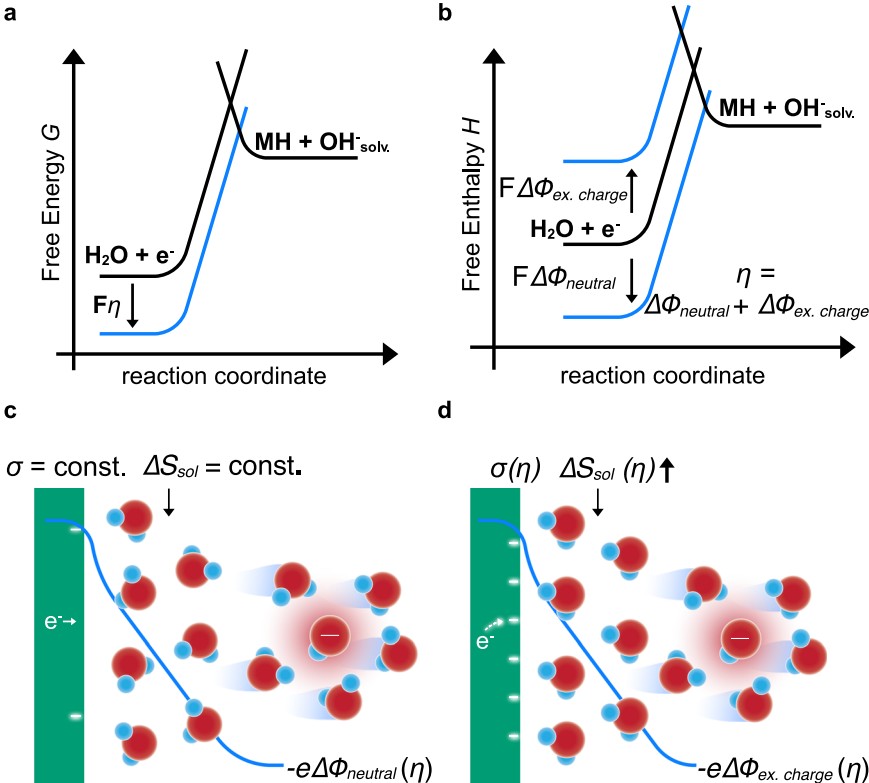

**Fig. 5 | Schematic of inner-sphere energetics. a** Simplified schematic of the free energy, *G*, for the water dissociation and hydroxide solvation reaction during the alkaline HER ($2H_2O + 2e^- \rightarrow H_2 + 2OH^-$), where the water molecule and electron are at the initial state and a metal hydride, $MH\cdot$ and solvated hydroxide, $OH^-_{solv}$ at the final state. Black lines represent the energetics at electrochemical equilibrium and the blue line under applied bias, where F is Faraday's constant and η the overpotential. **b** While *G* is reduced under bias, the free enthalpy, *H*, of the final state can in fact increase or decrease under bias (blue lines). For the conditions studied in Figs. 1–4, the bias leads to energetically reversible build-up of excess charge, which induces an electrostatic potential drop, $\Delta\phi_{ex.charge}$, and associated electric fields. These can impact the activation entropy in the solvent, $\Delta S_{sol}$. In contrast, Butler-Volmer theory assumes that the bias reduces $\Delta H$ by $\Delta\phi_{neutral}$ and that $\Delta S_{sol}$ is constant with bias. In this case, the electrochemical bias simply

manifests itself as an electrostatic potential difference, $\Delta\phi_{neutral}$. because the chemical potential differences are constant. Here, charge at the interface is either constant with bias or does not impact the kinetics. **c** With constant excess charge, σ, (and constant electric fields) the activation entropy, $\Delta S_{sol}$, stays constant and $\Delta\phi_{neutral}$ directly changes the enthalpy of the solvated ion, similar to outer-sphere reactions. Note, in these cases $\Delta\phi_{neutral}$ is the driving force and not an electric field. **d** Bias dependent build-up of excess charge at the interface leads to an electrostatic potential drop and electric fields that can increase the entropy of the solvation transition state, and, thus, the activation entropy, $\Delta S_{sol}$, and drive the reaction forward, i.e. reduce $\Delta G$ reaction in panel a, despite an increasing $\Delta H$ in panel b. In this case, the reaction is driven by entropic changes, in contrast to the enthalpic changes in panel c.

discharge that is only apparent for a more strongly charged surfaces. Full time resolution of the pH dependent kinetics are provided in Supplementary Movies 2–5. Taken together, the results in Fig. 4 indicate that the non-Nernstian overpotential shifts for the AOR are reflected in a pH dependent activation entropy. In line with our results above and the slope $d\log A dE_A^{-1}$ obtained here, we assign the changes in A primarily to entropic changes in the interfacial solvent. With lower pH, the activation entropy of the solvation step never reaches the maximum obtained for higher pH values, resulting in slower kinetics. Conversely, non-Nernstian overpotential shifts might be linked to the pH dependent formation of a charged intermediate, but to understand the effect on the kinetics, the activation entropy in the solvent and on the surface is as important as the activation enthalpy.

## Discussion

Traditional electrocatalyst research and education is largely based on the application of an outer-sphere theory to oversimplified, charge neutral model surfaces. Here, it is assumed that free energy changes (Fig. 5a) between the initial and final states are solely caused by a reduction of the free enthalpy (Fig. 5b) with bias. However, except for a few fast reactions and catalysts, most notably the HER in acid on Pt, most electrocatalyst surfaces are substantially polarized before the onset of Faradaic current, i.e. they display a significant kinetic

overpotential. The associated excess charge has been previously linked to an increasing activation enthalpy (Fig. 5b), based on experimental[18] and theoretical studies[1]. From these and other studies, a common notion developed that interfacial $H^+$ or $OH^-$ transport are suppressed in an ordered hydrogen-bond network. However, the effect of the bias on the activation entropy in the pre-exponential factor had not been addressed in these works, due to limitations of current computational work and the scarcity of temperature-dependent experimental data on the bias-dependence of the catalytic rates. In fact, the neglect of the latter in the literature has promoted the application of outer-sphere Butler-Volmer or Marcus-Hush-type theories that assume (largely without any experimental confirmation) a constant activation entropy with bias and rates that only increase due to a reduction of the activation enthalpy (Fig. 5c). These simplified models are often joined with oversimplified DFT leading to Volcano activity plots.

The results presented here and previously[16] indicate that an increasing activation entropy can (over)compensate the increasing activation enthalpy (Fig. 5d) and, thus, reduce the activation free energy (Fig. 5a). Previously, similar conclusions were drawn by Conway[7,8,25,26] for the HER on Hg from acidic proton donors, but its relevance for water dissociation and hydroxide solvation, let alone, other reactions has remained unclear. By applying a comprehensive

kinetic model, we conclude that an ordered hydrogen bond network can in fact (entropically) accelerate interfacial charge transfer, and not suppress it. This might be due to an effective delocalization of the ion in the transition state, and, thus, an increased transition state entropy, $S_t$. In other words, the number of accessible microstates in the transition state increases with bias, thus, increasing the probability of the barrier crossing event. At the same time, ordering of a reactant water molecule could also reduce the entropy of the initial state, $S_i$, which would additionally increase the activation entropy, $\Delta S = S_t - S_i$. This increase in the activation entropy comes at the (enthalpic) expense of increasing excess charge at the surface. We note, that the electrostatic potential drops in Fig. 5c, d are different. $\Delta\phi_{ex.charge}$ is linked to excess charge at the surface and can lead to electric fields that act over the spatially extended water network. In contrast, $\Delta\phi_{neutral}$ is a manifestation of the electrochemical bias due to constant chemical potentials[47], i.e. it occurs even in absence of charging. $\Delta\phi_{neutral}$ is the driving force (not an electric field) in Butler-Volmer theory, assuming a bias independent solvent and surface structure.

On statistical mechanics grounds, the observation of linear compensation slopes between the pre-exponential factor and the activation energy can be reconciled by the effect of the local environment on the shared bond-energy factors that impact the partition functions of, both $\Delta H$ and $\Delta S$[48]. Going beyond previous work by Conway[7,8,25,26], our results indicate a critical role of excess charge on the solvation kinetics during electrosorption, especially in a dynamically changing dipole network. These results build a bridge between fundamental electrochemistry insights extracted using cyclic voltammetry and electrocatalyst kinetics. Furthermore, compensation effects occur during the time dependent coverage of poisoning intermediates and need to be considered to understand non-Nernstian overpotential shifts with pH. Thus, understanding how the partition functions are linked and depend on the time and bias dependent interfacial environment and, in particular, the emergence of excess charge, constitutes one of the most important research topics in electro-, thermal and biocatalysis. This notion is also supported by operando spectroscopy studies that repeatedly detected charged (electron-poor) terminations during the oxygen evolution reaction (OER) on $IrO_X$[49–52] and $CoO_X$[52]. It has been argued[53–55] that incomplete (frustrated) phase transitions are important for nucleophilic attack by water molecules or hydroxides, similarly as for water oxidation in photosystem II[54,55]. Now, our results imply that such charged states are electrostatically coupled to the solvation kinetics. In fact, we hypothesize that water reactivity or hydroxide attack are fundamentally linked to the solvation kinetics and that the very existence of excess charge at solid-liquid interfaces might partially be caused by the need to (de)solvate ions in polar solvents.

Our results demonstrate the wealth of kinetic information that can be obtained via bias and temperature dependent studies. However, such experimental studies are more tedious than widespread cyclic voltammograms or Tafel "analysis" at a single temperature. They are more time consuming, require more care of providing contamination-free cells and long-term stable reference electrodes and accurate temperature control. They typically also require more durable catalyst performance. However, when combined with potential (or current) jump studies, dynamic catalyst kinetics can be resolved with unprecedented insight. Expanding such electrochemical studies and combining them with *operando* spectroscopy and microscopy and new computational approaches will enable to community to finally gain the knowledge needed to develop knowledge-driven catalyst design strategies.

## Methods

All the experiments were carried out in a classical three-electrode cell configuration. The polycrystalline platinum working electrode was prepared by cutting a known area from a Pt foil (50 µm, 99.995%, Fisher Scientific) and a platinum wire (0.5 mm, 99.995%, Fisher Scientific)

used as a counter electrode. The Pt foil working electrode was rinsed with 30% nitric acid, flame-annealed and cooled down with ultrapure water just before each experiment. A commercial RHE (ET096 mini Hydrogen-Reference Electrode eDAQ) or a leak-less Ag/AgCl electrode (eDAQ) were used as reference electrodes placed directly inside the working solution and ~1 cm from the working electrode. For the HER studies, the working solution was always deoxygenated by bubbling Ar for 15 min prior to measurement and keeping an Ar atmosphere during the entire time interval of the experiment. For the ORR, $O_2$ was bubbled continuously and an $O_2$ atmosphere maintained. A Gamry Reference 3000 potentiostat was used.

The impedance spectroscopy was carried out in a frequency range between 1 MHz and 1 Hz with 3 mV amplitude. For temperature dependent Arrhenius analysis, multi-step chronoamperometries were performed at temperatures between 10 and 30 °C in 5 °C intervals. For this purpose, the working electrode potential was held at pre-defined potentials for 15 s, while the current was recorded for the HER and ORR. The maximum temperature was limited to 10–25 °C for the ORR to prevent outgassing of the reactants at higher temperatures. In this case, the solubility of $O_2$ is 0.9-1.5 mmol/L[56]. Here, we focus on the onset potential region where the reaction is not affected by mass transport. At 25 °C, the current density range is ~30 µA cm$^{-2}$ to 1 mA cm$^{-2}$ (see e.g. Fig. 1b). Conversely, when we tested for the impact of IR compensation, we did not see any discernible change in the kinetic maps. Thus, all data is reported without IR correction. In the case of AOR, due to the problem of the catalyst deactivation by the formation of poisoning species (NO* and N*), chronoamperometry was performed by applying potential jumps between 0.1 V and AOR onset potentials. The potential at 0.1 V was held for 4 s to strip the poison species formed during the reaction and the AOR onset potentials were held for 2 s to record the time evolution of the rate with <1 ms resolution. For Arrhenius analysis, the $\log_{10}$ of the measured current densities were linearly fitted against 1/T [K$^{-1}$] using a least-squares linear regression. This is done to fit the data to the Arrhenius equation (J = A*exp (-$E_A$/(RT))) with the form $\log_{10}J = \log_{10}A - E_A/(2.3\,RT)$, where J is the measured current density for a given potential, A is the pre-exponential factor, $E_A$ the activation energy, R the ideal gas constant and T the temperature in Kelvin. Therefore, $E_A$ is obtained from the slope of the linear regression and the pre-exponential factor A from the origin intercept where 1/T tends to zero as seen in Supplementary Fig. 1b.

The electrochemical PTFE cell was kept in a secondary, double-walled bath which was temperature-controlled by a Julabo CORIO CP-300F and a Teflon-coated thermocouple inside the cell. For the ORR, constant bubbling of $O_2$ was needed to ensure saturation. To avoid interference of the bubbling during the impedance spectroscopy, a Teflon cylinder connected to the cell atmosphere was used to capture bubbles originating from the bottom of the solution.

All solutions were prepared using ultrapure water from an Elga PURELAB flex system (resistivity 18.2 MΩ cm and TOC < 1 ppb). 0.1 M KOH solutions were prepared using potassium hydroxide monohydrate (semiconductor grade, Sigma-Aldrich). 25% Ammonia solution (Merck) was employed to prepare the 0.1 M $NH_3$ solutions. The solutions at pH below 13 were prepared by adding the required amount of 70% $HClO_4$ (ACS reagent, Sigma-Aldrich) to a 0.1 M solution of KOH and $NH_3$. High purity (5–6 N) gases from Linde were used. Both the material of the electrochemical cell and the material used to prepare the solutions was Teflon. To clean it, it was kept in an acid solution of potassium permanganate before being treated with piranha solution and rinsed with ultrapure water.

To calculate an iso-current line (Figs. 1–2), the kinetic map is discretized in a grid, assigning to every point in space a pair of values of $\log_{10}A$ and $E_A$ and using the Arrhenius equation to calculate the current density of each point given a certain temperature of (25 °C in this work). Lastly, the computed current densities are grouped and

selected by value, connecting the points with the closest current density between them and plotted as a contour on top of the kinetic map. Note that the slope of the iso-current line corresponds to the variation of $\log_{10} A / E_A$ that results in a constant current density and that this value changes depending on the temperature used in their calculation, e.g. 0.175 mol kJ$^{-1}$ at 25 °C, 0.164 mol kJ$^{-1}$ at 45 °C and 0.154 mol kJ$^{-1}$ at 65 °C.

## Data availability

The authors declare that the data supporting the findings of this study are available within the paper and its Supplementary Information files. Additional data is available from the corresponding author upon reasonable request. Source data for Figs. 1–4 are provided with this paper. Source data are provided with this paper.

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

## Acknowledgements

F.S. acknowledges funding by BASF as part of the BasCat—UniCat BASF JointLab and the European Union's Horizon 2020 research and innovation program under the Marie Skłodowska-Curie grant agreement no. 101069017. C.G.R. acknowledges funding by the Deutsche Forschungsgemeinschaft (DFG, German Research Foundation)—505677835. S.Z.O. acknowledges funding of the European Union's Horizon (ERC, ORION, 101077895) and B.R.C. the funding of the German Federal Ministry of Education and Research (BMBF) under the grant Catlab (03EW0015B).

## Author contributions

S.Z.O. and B.R.C. conceived of the project. F.S. performed all experiments and C.G.R. computationally processed all data. F.S., C.G.R. and S.Z.O. analyzed the data and discussed them with B.R.C. F.S., C.G.R. and S.Z.O. wrote the manuscript with help of B.R.C.

## Funding

## Competing interests

The authors declare no competing interests.
