## [Peer Review File · Nature Communications]

Exploring Dynamic Solvation Kinetics at Electrocatalyst SurfacesEditorial Note: This manuscript has been previously reviewed at another journal that is not operating a transparent peer review scheme. This document only contains reviewer comments and rebuttal letters for versions considered at *Nature Communications*.

REVIEWER COMMENTS

Reviewer #1 (Remarks to the Author):

After reading the manuscript from Oener et al. as well as the correspondence from previous revision process, I would say following:

- 1) the key novelty is that different faradaic reactions exhibit similar slope of activation energy vs. activation entropy dependence. In my opinion, that is an interesting and valuable result that deserves attention. Despite, in their previous work in *Nature Energy*, they found similar for HER in alkaline media and water dissociation (electric field driven heterogeneous reaction), I believe that the result shown in this work is challenge for current understanding of catalytic reactions at electrified interfaces...
- 2) text of the manuscript and rebuttal indicate that authors have respectable competence. Maybe authors did not answer on every question of the referees in 100 % convincing manner, but generally they contemplated thoroughly about questions of the referees.
- 3) However, both referees stated that the manuscript, in some points, is difficult to read. I would disagree, it is much worse than that. For me there are many parts that I had a “headache” to understand what they meant to say.

So overall, this material has some valuable content, but it is written (in a best case) for some specialized physical chemistry or electrochemistry journal. Therefore, I would recommend transfer to more specialized journal or that authors consult with somebody who understands these matters and can help them to rewrite the manuscript completely, to be appropriate for journal with more general audience like *NatComm*.

Reviewer #2 (Remarks to the Author):

The authors have satisfactorily addressed my concerns, and the added/edited discussion, I believe, makes the text more clear. It is now fitting for publication.

Reviewer #3 (Remarks to the Author):

I co-reviewed this manuscript with one of the reviewers who provided the listed reports. This is part of the *Nature Communications* initiative to facilitate training in peer review and to provide appropriate recognition for Early Career Researchers who co-review manuscripts.

Reviewer #4 (Remarks to the Author):

The article ‘Dynamic Solvation Kinetics at Electrocatalyst Surfaces’ studies the interfacial solvation kinetics of electrochemical reactions, such as hydrogen evolution, ammonia oxidation, and oxygen

reduction. In the manuscript, the authors stress the importance of the non-equilibrium capacitance, pre-exponential factor (A), and activation energies (EA) to support interfacial ion solvation effects on electrocatalysis activity. The authors extracted the values of $\log A$ and EA from chronopotentiometries in the presence of excess charge, which revealed an increase in bias-dependent entropy and enthalpy. Furthermore, the authors utilized potential-jump experiments and impedance spectroscopy to investigate the correlation between pseudo-capacitive (dis)charge currents and faradaic kinetics and non-Nernstian kinetics and pH-dependent activation entropy of ammonia oxidation reaction. Overall, these findings discuss how excess charge leads to the water ordering in the transition state (e.g., increase in entropic change in the interfacial solvent) while penalizing the increase in the activation energy, which is not captured in Butler-Volmer theory. This manuscript does provide significant insight into the role of the entropy of the solvent, surface configurational entropy changes in electrocatalysis kinetics which makes it of value to the community. However, several technical issues need to be addressed before publication. These issues are detailed below:

1. The authors mentioned that EA and A were extracted from chronopotentiometry measurement. However, I cannot find enough experimental details on how those were obtained from the chronopotentiometry measurements. The authors need to add more detail on how EA and A were calculated.
2. The authors note that a bias modulates both the entropy and enthalpy of activation in electrocatalytic reactions. In simple models of electron transfer, however, the potential is generally described as modulating the enthalpic activation barrier of the reaction. However, some key questions remain. In particular, the authors should discuss in more detail through which mechanism the applied potential influences the activation entropy and activation enthalpy. For example: Is the activation entropy, such as water ordering, the driving force of electrocatalysis? How is the (Gibbs) free energy modulated by the applied potential? Does the change in activation entropy dominate the free energy of activation, even when the activation enthalpy increases with applied bias? Are cations and anions involved in the activated complex?
3. The authors state that increasing the overpotential leads to a higher interfacial entropy. The kinetically relevant parameter entering into the pre-exponential factor (which increases with overpotential), however, is the entropy of activation. The question thus is whether the formation of the activated complex involves an increase or decrease of the entropy of the system. This question is critical for understanding the role through which the potential acts on the reaction rate but is not sufficiently discussed in the manuscript. Detailed discussion of the entropy change upon formation of the transition state needs to be added to the manuscript.
4. The authors state that excess charge at the surface leads to an increase in the activation barrier. Why does excess charge cause a high enthalpy of activation? More discussion needs to be added.
5. In ORR, oxygen was introduced as a gaseous reactant. However, the solubility of oxygen in water decreases with increasing temperature, which should result in decreasing ORR rates. How did the authors account for this effect?
6. The authors stated that the slope of $d\log(A)/dEA$ is marked as hollow black symbols for HER on Page 3. However, the hollow black symbols are shown in Figure 1b, not Figure 2b.

We sincerely thank all reviewers for carefully evaluating our manuscript and for the thoughtful comments. We believe, they have helped us to improve the manuscript further. The comments of the Reviewers are given below in black text and our response to each comment is given in blue text. Changes/additions to the manuscript text are highlighted in yellow and italic font. In general, we have edited the text throughout the manuscript with the goal to make it more accessible to a wider audience, according to suggestions by Reviewer #1 and the editor. This included moving Fig. 1d-e and several text sections into the SI for the interested reader. A manuscript with the highlighted changes is attached.

Reviewer #1 (Remarks to the Author):

After reading the manuscript from Oener et al. as well as the correspondence from previous revision process, I would say following:

1) the key novelty is that different faradaic reactions exhibit similar slope of activation energy vs. activation entropy dependence. In my opinion, that is an interesting and valuable result that deserves attention. Despite, in their previous work in Nature Energy, they found similar for HER in alkaline media and water dissociation (electric field driven heterogeneous reaction), I believe that the result shown in this work is challenge for current understanding of catalytic reactions at electrified interfaces...

We sincerely thank the reviewer for the positive evaluation of our work and for highlighting the importance of our findings.

2) text of the manuscript and rebuttal indicate that authors have respectable competence. Maybe authors did not answer on every question of the referees in 100 % convincing manner, but generally they contemplated thoroughly about questions of the referees.

We thank the reviewer for the kind words.

3) However, both referees stated that the manuscript, in some points, is difficult to read. I would disagree, it is much worse than that. For me there are many parts that I had a "headache" to understand what they meant to say.

So overall, this material has some valuable content, but it is written (in a best case) for some specialized physical chemistry or electrochemistry journal. Therefore, I would recommend transfer to more specialized journal or that authors consult with somebody who understands these matters and can help them to rewrite the manuscript completely, to be appropriate for journal with more general audience like NatComm.

We apologize for not having been clearer. We understand that entropy – enthalpy compensation effects are a challenging topic. Compensation effects require us to simultaneously consider the effect of excess charge or coverage effects on the activation energy and entropy of the surface and solvent. We have now substantially edited the manuscript, trying to avoid over-specialized jargon and making the findings more accessible to a broader audience. We have also moved some sections about the capacitance and time constants to the SI for the more specialized readers. Due to the large amounts of edits that we have made throughout the manuscript in order to address the Reviewer's comments, we refrain from listing all edits here and kindly refer the reviewer to the manuscript file with the highlighted changes.

Reviewer #2 (Remarks to the Author):

The authors have satisfactorily addressed my concerns, and the added/edited discussion, I believe, makes the text more clear. It is now fitting for publication.

We sincerely thank the reviewer for the attention to detail and helping us to improve our manuscript substantially over the course of several reviews.

Reviewer #3 (Remarks to the Author):

We sincerely thank the reviewer for carefully evaluating our manuscript and appreciate the inclusion of Early Career Researchers in the review process.

Reviewer #4 (Remarks to the Author):

The article 'Dynamic Solvation Kinetics at Electrocatalyst Surfaces' studies the interfacial solvation kinetics of electrochemical reactions, such as hydrogen evolution, ammonia oxidation, and oxygen reduction. In the manuscript, the authors stress the importance of the non-equilibrium capacitance, pre-exponential factor (A), and activation energies (EA) to support interfacial ion solvation effects on electrocatalysis activity. The authors extracted the values of $\log A$ and EA from chronopotentiometries in the presence of excess charge, which revealed an increase in bias-dependent entropy and enthalpy. Furthermore, the authors utilized potential-jump experiments and impedance spectroscopy to investigate the correlation between pseudo-capacitive (dis)charge currents and faradaic kinetics and non-Nernstian kinetics and pH-dependent activation entropy of ammonia oxidation reaction. Overall, these findings discuss how excess charge leads to the water ordering in the transition state (e.g., increase in entropic change in the interfacial solvent) while penalizing the increase in the activation energy, which is not captured in Butler-Volmer theory.

This manuscript does provide significant insight into the role of the entropy of the solvent, surface configurational entropy changes in electrocatalysis kinetics which makes it of value to the community.

We sincerely thank the reviewer for the time dedicated to this review and the positive evaluation of our manuscript.

However, several technical issues need to be addressed before publication. These issues are detailed below:

1. The authors mentioned that EA and A were extracted from chronopotentiometry measurement. However, I cannot find enough experimental details on how those were obtained from the chronopotentiometry measurements. The authors need to add more detail on how EA and A were calculated.

We thank the reviewer for raising this important point and apologize for the missing information. This has been now added to the methods section:

Page 15:

The impedance spectroscopy was carried out in a frequency range between 1 MHz and 1 Hz with 3 mV amplitude. For **temperature dependent** Arrhenius analysis, multi-step chronoamperometries were performed at temperatures between 10 and 30°C in 5°C intervals. For this purpose, the working electrode potential **was held** at pre-defined potentials for 15 s, while the current was recorded for the HER and ORR. **The maximum temperature was limited to 10-25°C for the ORR to prevent outgassing of the reactants at higher temperatures. In this case, the solubility of O₂ is 0.9-1.5mmol/L⁵⁶. Here, we focus on the onset potential region where the reaction is not affected by mass transport. At 25°C, the current density range is ~ 30 μA cm⁻² to 1 mA cm⁻² (see e.g. Fig. 1b).** In the case of AOR, due to the problem of the catalyst deactivation by the formation of poisoning species (NO* and N*), chronoamperometry was performed by applying potential jumps between 0.1V and AOR onset potentials. The potential at 0.1V was held for 4 s to strip the poison species formed during the reaction and the AOR onset potentials were held for 2 s to record the time evolution of the rate with <1 ms resolution. **For Arrhenius analysis, the log₁₀ of the measured current densities were linearly fitted against 1/T [K⁻¹] using a least-squares linear regression. This is done to fit the data to the Arrhenius equation ($J = A \cdot \exp(-E_A/(RT))$) with the form $\log_{10}J = \log_{10}A - E_A/(2.3 RT)$, where J is the measured current density for a given potential, A is the pre-exponential factor, E_A the activation energy, R the ideal gas constant and T the temperature in Kelvin. Therefore, E_A is obtained from the slope of the linear regression and the pre-exponential factor A from the origin intercept where 1/T tends to zero as seen in Supplementary Figure 1b.**

Additionally, we included two panels to Supplementary Figure 1, showing in more detail how E_A and A are obtained:

SI Page 3:

Supplementary Figure 1 | Temperature dependent Arrhenius analysis of the current density. **a**, HER chronoamperometric measurements at different temperatures. **b**, Arrhenius analysis of HER data to obtain A and E_A from the fit of the \log_{10} of the measured current density against the inverse of the temperature in K^{-1} . E_A is obtained from the slope of the fit while A is obtained from the origin intercept extrapolating the regression to $1/T \rightarrow 0$. Note that R is the ideal gas constant defined as $8.314 J K^{-1} mol^{-1}$. **c**, Heatmap for R^2 values from linear Arrhenius fits for Figure 1. The R^2 value for all HER, ORR and AOR is generally well above 0.95, ensuring high accuracy of the reported E_A and A values. The colors in the table emphasize the r^2 (coefficient of determination) of the extracted A and E_A .

2. The authors note that a bias modulates both the entropy and enthalpy of activation in electrocatalytic reactions. In simple models of electron transfer, however, the potential is generally described as modulating the enthalpic activation barrier of the reaction. However, some key questions remain. In particular, the authors should discuss in more detail through which mechanism the applied potential influences the activation entropy and activation enthalpy.

We sincerely thank the reviewer for the thoughtful questions that led us to expand the *Conclusion* section, which we have added to the rebuttal letter after question #4. Directly below, you find a point-by-point response to your questions.

For example: Is the activation entropy, such as water ordering, the driving force of electrocatalysis? How is the (Gibbs) free energy modulated by the applied potential? Does the change in activation entropy dominate the free energy of activation, even when the activation enthalpy increases with applied bias?

We thank the reviewer for raising these key questions. As indicated in Fig. 5a (see also below question #4), the Gibbs free energy is reduced by the bias to obtain increasing rates. However, this does not mean that the driving force for the reaction must be of enthalpic origin, as generally assumed. Here, we observe that the reaction is driven forward by the increasing activation entropy, which (over)compensates the increasing activation enthalpy. This is now discussed more clearly in the *Conclusions* (see below).

Are cations and anions involved in the activated complex?

This is an important question, which we had already discussed on Page 7. We have now added more explanation to this topic.

Page 7-8:

Electrolyte ions are a critical in electrochemistry^{33,34} and have been extensively shown to impact interfacial capacitance and catalyst kinetics. Previously, it has been hypothesized that anion adsorption during the HER in acid could lead to increasing pre-exponential factors, because it could draw the charged protons to the surface and, thus, increase the reactant concentration⁴. However, for the alkaline HER kinetics in Fig. 1, the reactant is neutral water and we observe a compensation slope of $\Delta \log A(\eta) \cdot \Delta E_A(\eta)^{-1} \sim 0.22 \text{ mol kJ}^{-1}$, which is similar to the ones we observed for pure water humidified bipolar membranes $(0.25 \text{ mol kJ}^{-1})^{16}$ in absence of auxiliary electrolyte. Therefore, we assign the entropic changes in Fig. 1-2 primarily to the intrinsic nature of interfacial water dipole ordering, but suspect that electrolyte ions can modify ΔS and ΔH . This will especially be true, if electrolyte ions adsorb to the surface, impact the hydrogen bond network or are needed to form the activated complex, as hypothesized for CO₂ reduction³⁵.

3. The authors state that increasing the overpotential leads to a higher interfacial entropy. The kinetically relevant parameter entering into the pre-exponential factor (which increases with overpotential), however, is the entropy of activation. The question thus is whether the formation of the activated complex involves an increase or decrease of the entropy of the system. This question is critical for understanding the role through which the potential acts on the reaction rate but is not sufficiently discussed in the manuscript. Detailed discussion of the entropy change upon formation of the transition state needs to be added to the manuscript.

We thank the reviewer for pointing out the needed clarification. We hypothesize, that the bias can increase the transition state entropy, S_t , due to an increasing delocalization of the ion in the ordered hydrogen bond network. At the same time, the entropy of the reactant state, S_i , which reflects the different degrees of freedom of the water molecule, could decrease with bias and increasing order. In both cases, the activation entropy, $\Delta S = S_t - S_i$, increases. This is now discussed more clearly in the *Conclusions* section (see below).

4. The authors state that excess charge at the surface leads to an increase in the activation barrier. Why does excess charge cause a high enthalpy of activation? More discussion needs to be added.

We thank the reviewer for pointing out the needed of clarification. Increasing activation energies with excess charge have been previously unveiled by MD simulations (<https://www.sciencedirect.com/science/article/pii/S1572665724000201>) and inferred from experiments (<https://www.nature.com/articles/nenergy201731>). In these works, the effect of excess charge on the activation enthalpy has been explained the effect of ordering of the interfacial hydrogen bonding network. As also noted by us, the effect of the bond-energy factors that determine the activation enthalpy (and activation entropy for compensation effects) in the statistical mechanics framework requires the calculation of the partition functions. This is beyond the scope of this work and will likely require more theoretical studies. This is now discussed more clearly in the *Conclusions* section (see below).

Page 12-13: **Conclusions**

Traditional electrocatalyst research and education is largely based on the application of an outer-sphere theory to oversimplified, charge neutral model surfaces. Here, it is assumed that free energy changes (Fig. 5a) between the initial and final states are solely caused by a reduction of the free enthalpy (Fig. 5b) with bias. However, except for a few fast reactions and catalysts, most notably the HER in acid on Pt, most electrocatalyst surfaces are substantially polarized before the onset of Faradaic current, i.e. they display a significant kinetic overpotential. The associated excess charge has been previously linked to an increasing activation enthalpy (Fig. 5b), based on experimental¹⁸ and theoretical studies¹. From these and other studies, a common notion developed that interfacial H^+ or OH^- transport are suppressed in an ordered hydrogen-bond network. However, the effect of the bias on the activation entropy in the pre-exponential factor had not been addressed in these works, due to limitations of current computational work and the scarcity of temperature-dependent experimental data on the bias-dependence of the catalytic rates. In fact, the neglect of the latter in the literature has promoted the application of outer-sphere Butler-Volmer or Marcus-Hush-type theories that assume (without any experimental confirmation) a constant activation entropy with bias and rates that only increase due to a reduction of the activation enthalpy (Fig. 5c). These simplified models are often joined with oversimplified DFT-based theory leading to Volcano activity plots.

Previous results by Conway^{7,8,25,26} and our own¹⁶ work indicate that an increasing activation entropy can (over)compensate the increasing activation enthalpy (Fig. 5d) and, thus, reduce the activation free energy (Fig. 5a). In other words, an ordered hydrogen bond network might be able to (entropically) accelerate interfacial charge

transfer. This might be due to an effective delocalization of the ion in the transition state, and, thus, an increased transition state entropy, S_t . At the same time, ordering of a reactant water molecule could also reduce the entropy of the initial state, S_i , which would additionally increase the activation entropy, $\Delta S = S_t - S_i$. This increase in the activation entropy comes at the (enthalpic) expense of increasing excess charge at the surface. We note, that the electrostatic potential drops in Fig. 5c and Fig 5d are different. $\Delta\phi_{ex.charge}$ is linked to excess charge at the surface and can lead to electric fields that act over the spatially extended water network. In contrast, $\Delta\phi_{neutral}$, is a manifestation of the electrochemical bias due to constant chemical potentials⁴⁷, i.e. it occurs even in absence of charging. $\Delta\phi_{neutral}$ is the driving force (not an electric field) in simple Butler-Volmer theory, assuming a bias independent solvent and surface structure.

On statistical mechanics grounds, the observation of linear compensation slopes between the pre-exponential factor and the activation energy can be reconciled by the effect of the local environment on the shared bond-energy factors that impact the partition functions of, both ΔH and ΔS ⁴⁸. Going beyond previous work by Conway^{7,8,25,26}, our results indicate a critical role of excess charge on the solvation kinetics during electrosorption, especially in a dynamically changing dipole network. These results build a bridge between fundamental electrochemistry insights extracted using cyclic voltammetry and electrocatalyst kinetics. Furthermore, compensation effects occur during the time dependent coverage of poisoning intermediates and need to be considered to understand non-Nernstian overpotential shifts with pH. Thus, understanding how the partition functions are linked and depend on the time and bias dependent interfacial environment and, in particular, the emergence of excess charge, constitutes one of the most important research topics in electro-, thermal and biocatalysis. This notion is also supported by operando spectroscopy studies that repeatedly detected charged (electron-poor) terminations during the oxygen evolution reaction (OER) on IrO_x ⁴⁹⁻⁵² and CoO_x ⁵². It has been argued⁵³⁻⁵⁵ that incomplete (frustrated) phase transitions are important for nucleophilic attack by water molecules or hydroxides, similarly as for water oxidation in photosystem II^{54,55}. Now, our results imply that such charged states are electrostatically coupled to the solvation kinetics. In fact, we hypothesize that water reactivity or hydroxide attack are fundamentally linked to the solvation kinetics and that the very existence of excess charge at solid-liquid interfaces might partially be caused by the need to (de)solvate ions in polar solvents.

Fig. 5 | Schematic of inner-sphere energetics. **a**, Simplified schematic of the free energy, G , for the water dissociation and hydroxide solvation reaction during the alkaline HER ($2H_2O + 2e^- \rightarrow H_2 + 2OH^-$), where the water molecule and electron are at the initial state and a metal hydride, MH , and solvated hydroxide, OH^-_{solv} at the final state. Black lines represent the energetics at electrochemical equilibrium and the blue line under applied bias, where F is Faraday's constant and η the overpotential. **b**, While G is reduced under bias, the free enthalpy, H , of the final state can in fact increase or decrease under bias (blue lines). For the conditions studied in Fig. 1-4, the bias leads to energetically reversible build-up of excess charge, which induces an electrostatic potential drop, $\Delta\phi_{ex. charge}$, and associated electric fields. These can impact the activation entropy, ΔS . In contrast, Butler-Volmer theory assumes that the bias energetically irreversibly reduces ΔH by $\Delta\phi_{neutral}$ and that ΔS is bias independent. In this case, the electrochemical bias simply manifests itself as an electrostatic potential difference, $\Delta\phi_{neutral}$, because the chemical potential differences are constant. Here, charge at the interface is either constant with bias or does not impact the kinetics. **c**, With constant excess charge, σ , (and constant electric fields) the activation entropy, ΔS , stays constant and $\Delta\phi_{neutral}$ directly changes the enthalpy of the solvated ion, similar to outer-sphere reactions. Note, in these cases $\Delta\phi_{neutral}$ is the driving force and not an electric field. **d**, Bias dependent build-up of excess charge at the interface leads to an electrostatic potential drop and electric fields that can increase **the entropy of the solvation transition state, and, thus, the activation entropy, ΔS** , and drive the reaction forward, i.e. reduce ΔG reaction in panel a, despite an increasing ΔH in panel b. In this case, the reaction is purely driven by entropic changes, in contrast to the enthalpic changes in panel c.

5. In ORR, oxygen was introduced as a gaseous reactant. However, the solubility of oxygen in water decreases with increasing temperature, which should result in decreasing ORR rates. How did the authors account for this effect?

This is a good observation. In fact, we had observed decreasing rates with increasing temperatures for temperatures above $\sim 35^\circ\text{C}$ because of the outgassing of O_2 from the electrolyte. This led to a thermodynamic shift of the onset potential, lowered the mass-transport and led to low R^2 values for the linear Arrhenius fits. Therefore, we chose to restrict the analysis for the ORR to $5\text{-}25^\circ\text{C}$ for which we observed increasing rates with increasing temperatures and obtained high R^2 values (Supplementary Figure 1). We added the following explanation and description to the Methods:

Page 15:

The impedance spectroscopy was carried out in a frequency range between 1 MHz and 1 Hz with 3 mV amplitude. For temperature dependent Arrhenius analysis, multi-step chronoamperometries were performed at temperatures between 10 and 30°C in 5°C intervals. For this purpose, the working electrode potential was held at pre-defined potentials for 15 s, while the current was recorded for the HER and ORR. The maximum temperature was limited to $10\text{-}25^\circ\text{C}$ for the ORR to prevent outgassing of the reactants at higher temperatures. In this case, the solubility of O_2 is $0.9\text{-}1.5\text{mmol/L}$ ⁵⁶. Here, we focus on the onset potential region where the reaction is not affected by mass transport. At 25°C , the current density range is $\sim 30\ \mu\text{A cm}^{-2}$ to $1\ \text{mA cm}^{-2}$ (see e.g. Fig. 1b). In the case of AOR, due to the problem of the catalyst deactivation by the formation of poisoning species (NO^ and N^*), chronoamperometry was performed by applying potential jumps between 0.1V and AOR onset potentials. The potential at 0.1V was held for 4 s to strip the poison species formed during the reaction and the AOR onset potentials were held for 2 s to record the time evolution of the rate with $<1\ \text{ms}$ resolution. For Arrhenius analysis, the \log_{10} of the measured current densities were linearly fitted against $1/T\ [\text{K}^{-1}]$ using a least-squares linear regression. This is done to fit the data to the Arrhenius equation ($J = A \cdot \exp(-E_A/(RT))$) with the form $\log_{10}J = \log_{10}A - E_A/(2.3 RT)$, where J is the measured current density for a given potential, A is the pre-exponential factor, E_A the activation energy, R the ideal gas constant and T the temperature in Kelvin. Therefore, E_A is obtained from the slope of the linear regression and the pre-exponential factor A from the origin intercept where $1/T$ tends to zero as seen in Supplementary Figure 1b.*

[56] Xing, W., Yin, M., Lv, Q., Hu, Y., Liu, C., Zhang, J. Diffusion Coefficient, and Solution Viscosity. Rotating Electrode Methods and Oxygen Reduction Electrocatalysts. Xing, W., Yin, G., Zhang, J., Eds.; Elsevier: Amsterdam, 2014; pp 1-31.

6. The authors stated that the slope of $d\log(A)/dE_A$ is marked as hollow black symbols for HER on Page 3. However, the hollow black symbols are shown in Figure 1b, not Figure 2b.

We thank the reviewer for the attention to detail and apologize for our oversight. The main text has been corrected to reference Figure 1B and not Figure 2B.

Reviewer #1 (Remarks to the Author):

Authors addressed my comments properly.

Therefore, I support this valuable piece of work to be published in Nature Communication.

Reviewer #3 (Remarks to the Author):

Reviewer #4 (Remarks to the Author):

The authors have answered and addressed all my concerns in the revised manuscript.